# Methodological characteristics of Peruvian clinical practice guidelines, 2018 – 2023: A scoping review

Ana Brañez-Condorena[1], Blanca Solis-Chimoy[2], Jhonatan R. Mejia[3], Lesly Chávez-Rimache[4], David R. Soriano-Moreno[5], Jose Ernesto Fernández-Chinguel[6], Alvaro Taype-Rondan[1,4]*

1 EviSalud - Evidencias en Salud, Lima, Peru, 2 ADIECS Asociación para el Desarrollo de la Investigación Estudiantil en Ciencias de la Salud, Universidad Nacional Mayor de San Marcos, Lima, Peru, 3 Carrera de Medicina Humana, Universidad Científica del Sur, Lima, Peru, 4 Unidad de Investigación para la Generación y Síntesis de Evidencias en Salud, Vicerrectorado de Investigación, Universidad San Ignacio de Loyola, Lima, Peru, 5 Unidad de Investigación Clínica y Epidemiológica, Escuela de Medicina, Universidad Peruana Unión, Lima, Peru, 6 Facultad de Salud Pública y Administración, Universidad Peruana Cayetano Heredia, Lima, Peru

* alvaro.taype.r@gmail.com

## Abstract

### Background

Clinical practice guidelines (CPGs) must be developed through a rigorous and transparent methodology to ensure the appropriateness and reliability of their recommendations. In Peru, little is known regarding how CPGs adhere to established methodological standards.

### Objective

To describe the methodological characteristics of CPGs developed and published in Peru between 2018 and 2023.

### Methods

We conducted a scoping review and searched CPGs on Google, Google Scholar, and relevant local organizational websites. To be included, CPGs had to self-identify as such, have full-text versions available online, provide explicit methodological descriptions, and base their recommendations on systematic reviews, with publication dates between January 2018 and December 2023. We presented the results descriptively and analyzed the methodological differences among CPGs from different organizations using Fisher's exact tests.

**Data availability statement:** All relevant data are within the manuscript and its Supporting information files.

**Funding:** The author(s) received no specific funding for this work.

**Competing interests:** ABC, JRM, LCR and ATR have participated in the development of clinical practice guidelines for the Institute of Health Technology Assessment and Research (IETSI) of EsSalud and have received compensation for their work. The authors declare no additional potential conflicts of interest related to this study. This does not alter our adherence to PLOS ONE policies on sharing data and materials.

## Results

Out of 312 records assessed, 88 CPGs met the inclusion criteria. We found a declining publication trend over the study period: 39 CPGs were published in 2018–2019, 30 in 2020–2021, and 19 in 2022–2023. Most CPGs (60.2%) were developed by the Peruvian Social Security Health Insurance (EsSalud). Oncology was the most prevalent specialty of the CPGs (20.5%) and most CPGs (96.6%) included disease management. 23.7% of CPGs used ≥ 3 search engines, and 76.3% showed the risk of bias assessment. Although most CPGs indicated using the Grading of Recommendations, Assessment, Development, and Evaluations (GRADE) methodology (94.3%), 30.7% missed Summary of Findings tables, 38.6% did not include Evidence-to-Decision frameworks, and only 5.7% used minimal important difference for at least one question. Additionally, economic analyses were infrequently sought or included.

## Conclusions

This study highlights significant methodological deficiencies in Peruvian CPGs, including inadequate reporting of search strategies, bias assessments, and key GRADE components. Addressing these shortcomings is crucial for enhancing the quality and reliability of CPGs and promoting equitable healthcare delivery in Peru.

---

## Introduction

Evidence-based clinical practice guidelines (CPGs) are defined by the Institute of Medicine as "statements that include recommendations, intended to optimize patient care, informed by a systematic review of evidence and an assessment of the benefits and harms of alternative care options" [1]. To ensure these recommendations are optimal, CPGs must be developed using a rigorous, transparent, and effective methodology [2,3]. A trustworthy CPG adheres to these standards that emphasize key components, including clearly defined clinical questions, systematic literature searches, the use of a grading system for evidence, explicit criteria for formulating recommendations, external review, and a structured process for updates [1,2].

Local CPGs, developed by countries, health insurance entities, or hospitals, provide valuable context-specific recommendations by considering context-dependent factors such as costs, feasibility, equity, and acceptability. However, previous studies evaluating local CPGs in Latin American countries, including Brazil [4], Mexico [5], and Chile [6], have identified significant methodological deficiencies. Common shortcomings include unclear processes for formulating recommendations [4–6], inadequate systematic searches and selection criteria [4,5], limited consideration of evidence strengths and limitations [4–6], and failure to incorporate patients' preferences [4,6]. In Peru, an evaluation of 17 CPGs published by the Ministry of Health between 2009 and 2014 revealed similar deficiencies [7], including a lack of information about the guideline development team, minimal use of evidence, and unclear recommendation processes [7]. A further study of 31 CPGs developed by the

Peruvian Social Health Insurance System (EsSalud) up to December 2015 reported gaps in evidence-based frameworks and inconsistent reporting of recommendations [8].

In response, the Peruvian Ministry of Health published a technical document in 2015 standardizing CPG methodology, which is mandatory in the public sector and recommended for the private sector. This document stipulated that CPGs should formulate clinical questions and conduct systematic searches to answer them. It also mandated the use of the Grading of Recommendations, Assessment, Development, and Evaluations (GRADE) methodology to assess the certainty of the evidence, create Summary of Findings (SoF) tables, and formulate strong or conditional recommendations [9]. Despite the existence of this standard, CPG development in Peru remains decentralized. Different institutions, even within the same sector, develop CPGs independently, with distinct methodological teams and varying capacities. This institutional heterogeneity can influence the methodological quality and consistency of CPGs across the country.

Internationally, institutions such as the World Health Organization (WHO) [10], National Institute for Health and Care Excellence (NICE) in the UK [11], and regional organizations such as the Ministry of Health of Argentina [12] and Chile [13] follow rigorous methodologies emphasizing transparency, evidence quality assessment, and regular updates. While Peru's standards have advanced, they could benefit from greater alignment with these international practices, ensuring guidelines remain evidence-based and relevant.

No studies have evaluated whether Peruvian CPGs published after the release of this technical document comply with the globally standardized methodological characteristics required by national regulations. Therefore, we conducted this study to describe the methodological characteristics of the CPGs developed in Peru and published between 2018 and 2023.

## Materials and methods

We performed a scoping review following the Preferred Reporting Items for Systematic Reviews and Meta-Analyses for Scoping Reviews (PRISMA-ScR) guidelines [14].

### Eligibility criteria

We included CPGs developed by public or private organizations in Peru that fulfilled the following criteria: 1) Documents self-named as CPGs, 2) CPGs with full-text versions publicly available online, 3) CPGs that explained -at least briefly- the methodology followed for its development, 4) CPGs that provided recommendations and indicated that these recommendations were based on systematic reviews (SRs), and 5) CPGs published between January 2018 and December 2023 (when the CPG did not present a publication date, the approval resolution date was considered).

We included Peruvian CPGs that reported basing their recommendations on SRs, an essential component of evidence-based CPG development. This approach allowed us to focus on CPGs with an evidence-based intent and assess whether they incorporated key methodological components aligned with international standards [10–13] and the national regulations for CPG development in Peru [9].

CPGs that reported only partial updates (revisions and modifications of specific sections to incorporate new evidence, recommendations update, minor methodologies adjusts, or reflect regulatory changes without a full overhaul) were excluded. No restriction was applied regarding the level of healthcare (primary, secondary, or tertiary) for which the CPGs were intended.

### Search and selection of CPG

Two authors (ABC and BSC) independently searched for CPGs on Google and Google Scholar, and the websites of local organizations likely to develop CPGs (S1 Table). Regarding Google and Google Scholar, CPGs were searched until the results yielded at least 100 consecutive entries that did not meet the predefined search criteria. If CPGs from other local organizations were found through Google and Google Scholar, additional searches for other CPGs were conducted on the websites of those organizations. The search was performed on March 22, 2024.

Duplicate CPGs were removed manually. Two authors (ABC and BSC) independently selected the CPGs that met the eligibility criteria for inclusion. In case of discrepancies, a third investigator (ATR) was consulted.

## Data extraction

When a CPG was found, all its versions and documents (short version, summarized version, article version, full-text, supplementary material, and others) were searched on Google and the website of the organization responsible for its development.

A pilot study was conducted to standardize the data extraction process. Subsequently, six investigators (ABC, BSC, JRM, DRSM, LCR, JEFC) worked in pairs to independently extract the variables of interest using a standardized Microsoft Excel sheet. In case of disagreements, consensus was reached through discussion among the investigators, followed by a second review of the data from the included CPGs to identify the precise information. To ensure the quality of the data collection process, a predefined set of criteria was applied to each variable. This included cross-checking the extracted information with the original documents, verifying consistency across multiple versions and journal articles of the CPGs, and ensuring the accuracy of all recorded data. If consensus could not be reached, another researcher (ATR) was consulted to resolve any uncertainties or clarify discrepancies in the data.

Variables of interest included the year of publication, covered area, developing organization (given that Peruvian health institutions operate with distinct processes and capacities), type of CPG ("de novo", adapted, or adopted), funding sources, targeted population, type and number of authors, patient flowchart report, questions, and table of recommendations report, type of recommendations, authors' conflict of interest statement, presence of external reviewers, article version, number of search engines, search strategies report, risk of bias assessment, use of a minimally important difference (MID), use of the GRADE methodology, SoF table report, GRADE Evidence to Decision (EtD) frameworks report, use of resources assessment, and cost-effectiveness assessment.

CPGs were classified according to the developing institution, rather than by broader health system sectors, to more accurately reflect the decentralized structure and variability of CPG development in Peru.

## Statistical analyses

Statistical analyses were conducted using Stata v.17 software. Categorical variables were presented as both absolute and relative frequencies, and numerical variables were summarized using measures of central tendency and dispersion. Additionally, Fisher's exact tests were performed to assess differences among the methodological characteristics of the CPGs across the developing organizations. A p-value $< 0.05$ was considered statistically significant.

## Results

We assessed 312 records, from which 88 CPGs fulfilled the inclusion criteria (Fig 1 and S2 Table). The number of published CPGs has shown a decreasing trend over time: 39 in 2018–2019, 30 in 2020–2021, and 19 in 2022–2023 (Fig 2).

Most of the included CPGs were developed by the Institute of Health Technology Assessment and Research (IETSI, for its initials in Spanish) of the Peruvian Social Security Health Insurance (EsSalud) (60.2%), followed by hospitals of the Ministry of Health of Peru (17.1%), the AUNA Peru Network (a private healthcare network primarily focused on oncology) (9.1%), the National Institute of Health (INS, for its initials in Spanish) of the Ministry of Health (9.1%), the General Directorate of Strategic Health Interventions (DGIESP, for its initials in Spanish) of the Ministry of Health (2.3%), and the Peruvian Society of Neurology (a private professional organization dedicated to neurology) (2.3%).

The most common covered area among the CPGs was oncology (20.5%), and most of the CPGs (96.6%) included disease management. Regarding authorship, the identification of authors and their respective roles were detailed in 97.7% and 92% of the CPGs, respectively. Among CPGs that detailed the author's roles, 100% reported having at least one clinical expert, 97.5% included at least one CPG development methodologist, and 3.7% included a patient representative. The

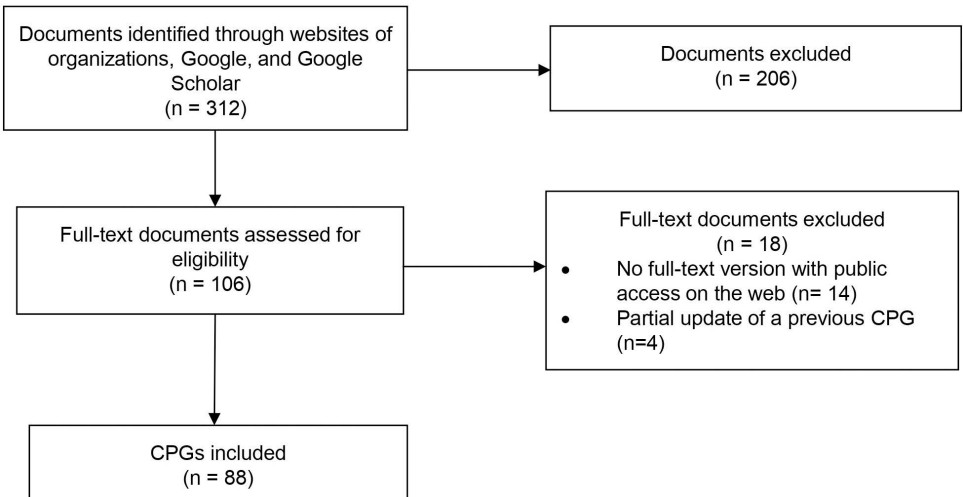

**Fig 1. Selection flowchart.** CPG: clinical practice guideline.

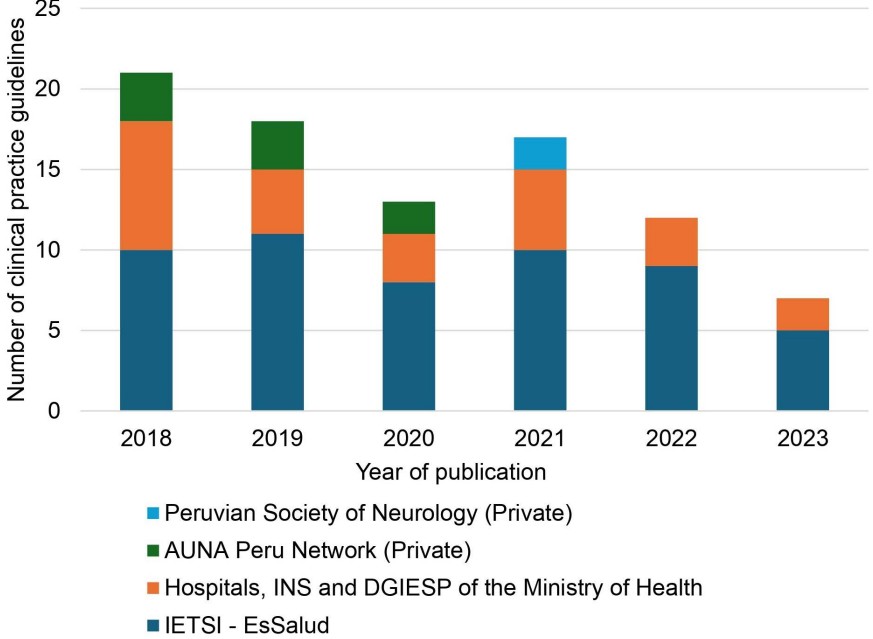

**Fig 2. Trends in the publication of clinical practice guidelines (2018-2023), per local organizations.** IETSI - EsSalud: Institute of Health Technology Assessment and Research of the Social Security Health Insurance; INS: National Institute of Health of the Ministry of Health; DGIESP: General Directorate of Strategic Health Interventions of the Ministry of Health.

median total number of authors was 13.5 (interquartile range [IQR]: 11–18), while the median numbers of clinical experts and methodologists were 9 (IQR: 6–12) and 3 (IQR: 2–5), respectively (Table 1).

Clinical flowcharts were included in 87.5% of CPGs, and 85.2% clearly outlined the clinical questions addressed, with a median of seven (IQR: 6–9) questions per CPG. Additionally, 94.3% of the CPGs differentiated between strong

**Table 1. General characteristics of Peruvian clinical practice guidelines (n = 88).**

| Characteristics [a] | n (%) |
|---|---|
| Local organization that developed the CPG | |
| IETSI – EsSalud | 53 (60.2) |
| Hospitals of the Ministry of Health | 15 (17.1) |
| AUNA Peru Network [b] | 8 (9.1) |
| INS – Ministry of Health | 8 (9.1) |
| DGIESP – Ministry of Health | 2 (2.3) |
| Peruvian Society of Neurology [b] | 2 (2.3) |
| Year of publication | |
| 2018–2019 | 39 (44.3) |
| 2020–2021 | 30 (34.1) |
| 2022–2023 | 19 (21.6) |
| Type of CPG [c] | |
| De novo | 64 (72.7) |
| Adapted from previous CPGs | 28 (31.8) |
| Adopted from previous CPGs | 12 (13.6) |
| Area addressed | |
| Oncology | 18 (20.5) |
| Ophthalmology | 11 (12.5) |
| Cardiology | 7 (8.0) |
| Neurology | 7 (8.0) |
| Others | 45 (51.1) |
| Target population | |
| Adults | 61 (69.3) |
| Children | 10 (11.4) |
| Adults and children | 7 (8.0) |
| Not specified | 10 (11.4) |
| Scenario addressed [c] | |
| Prevention | 21 (23.9) |
| Diagnosis | 52 (59.1) |
| Management | 85 (96.6) |
| Follow-up | 15 (17.1) |
| Authorship [c] | |
| The identification of authors was detailed | 86 (97.7) |
| The roles of the authors were detailed | 81 (92.0) |
| Role of authors [c] | |
| At least one author was a clinical expert (n = 81) | 81 (100.0) |
| At least one author was a methodologist (n = 81) | 79 (97.5) |
| At least one author was a patient (n = 81) | 3 (3.7) |
| At least one author had another role (n = 81) | 63 (77.8) |
| Number of authors [d] | |
| Total authors (n = 86) | 13.5 (11 - 18) |
| Clinical experts (n = 78) | 9 (6 - 12) |
| Methodologists (n = 76) | 3 (2 - 5) |
| Patients (n = 3) | 6 (2 - 7) |
| Others (n = 59) | 1 (1 - 2) |

*(Continued)*

**Table 1.** (Continued)

IETSI – EsSalud: Institute of Health Technology Assessment and Research of the Social Security Health Insurance; INS: National Institute of Health of the Ministry of Health; DGIESP: General Directorate of Strategic Health Interventions of the Ministry of Health.

[a]For the variables where information could not be obtained from all the CPGs, the number of CPGs from which information was available is indicated in parentheses (n=).

[b]Private organizations.

[c]The categories were not mutually exclusive.

[d]Median (Interquartile range).

recommendations (median: 4, IQR: 2–8), conditional recommendations (median: 4, IQR: 2–8), and good clinical practices (median: 14, IQR: 4–23). External reviewers assessed 89.8% of the CPGs. While no CPG reported receiving industry funding, 12.5% did not specify their funding sources. Furthermore, 9.1% did not disclose whether the authors had any conflicts of interest ([Table 2]).

Regarding the methodological characteristics of the CPGs, 23.7% employed three or more search engines, with higher frequencies observed in CPGs from the INS-Ministry of Health and AUNA Peru Network. Additionally, 88.2% of the CPGs

**Table 2.** Specific characteristics of Peruvian clinical practice guidelines (n = 88).

| Characteristics[a] | n (%) |
|---|---|
| Included clinical flowchart | 77 (87.5) |
| Clearly identifies the assessed clinical questions | 75 (85.2) |
| Number of clinical questions[b] | |
| Total clinical questions (n = 75) | 7 (6 - 9) |
| Intervention questions (n = 75) | 5 (4 - 7) |
| Diagnostic questions (n = 75) | 1 (0 - 3) |
| Show a table with all the recommendations | 82 (93.2) |
| Clear differentiation between strong recommendations, conditional recommendations, and good clinical practices | 83 (94.3) |
| Number of recommendations based on the GRADE methodology[b] | |
| Strong recommendations (n = 83) | 4 (2 - 8) |
| Conditional recommendations (n = 83) | 4 (2 - 8) |
| Good clinical practices (n = 83) | 14 (4 - 23) |
| Funding | |
| Not specified | 11 (12.5) |
| Declared funding but not from the industry | 77 (87.5) |
| Declared industry funding | 0 (0.0) |
| Declaration of author's conflicts of interest | |
| Did not declare author's conflicts of interest | 8 (9.1) |
| Declared that no author had conflicts of interest | 70 (79.5) |
| Declared that some authors had conflicts of interest | 10 (11.4) |
| Identification of external reviewers | 79 (89.8) |

CPG: Clinical practice guideline: GRADE: Grading of Recommendations, Assessment, Development, and Evaluations.

[a]For the variables where information could not be obtained from all the CPGs, the number of CPGs from which information was available is indicated in parentheses (n=).

[b]Median (Interquartilic range).

presented their search strategies, with IETSI-EsSalud and the INS-Ministry of Health achieving 100% adherence to this practice. The reporting of risk of bias assessments varied across organizations, ranging from 0% to 100% of the CPGs (Table 3).

Of the included CPGs, 94.3% indicated using the GRADE methodology, but only 69.3% presented SoF tables, with variations ranging from 0% to 88.7% across organizations. Only 5.7% of the CPGs established minimally important differences for interpreting the evidence, and this practice was exclusively observed in IETSI-EsSalud. Regarding the EtD framework, 38.6% did not use this framework, with lower frequencies observed in IETSI-EsSalud (22.6%) and the INS-Ministry of Health (25.0%). IETSI-EsSalud was the only organization that sought economic evidence in three CPGs (3.4%) and conducted an economic evaluation in one CPG (1.1%) (Table 3).

**Table 3. Methodological characteristics of Peruvian clinical practice guidelines (n = 88).**

| Characteristics[a] | Total | IETSI – EsSalud | Hospitals of the Ministry of Health | AUNA Peru Network | INS – Ministry of Health | Others[b] | p-value[c] |
|---|---|---|---|---|---|---|---|
| | n (%) | n (%) | n (%) | n (%) | n (%) | n (%) | |
| Number of search engines used to answer each clinical question (n = 76)[d] | | | | | | | **<0.001** |
| 1 | 23 (30.3) | 22 (44.0) | 0 (0.0) | 1 (14.3) | 0 (0.0) | 0 (0.0) | |
| 2 | 29 (38.2) | 25 (50.0) | 3 (37.5) | 0 (0.0) | 1 (12.5) | 0 (0.0) | |
| ≥ 3 | 18 (23.7) | 3 (6.0) | 2 (25.0) | 4 (57.1) | 7 (87.5) | 2 (66.7) | |
| Not mentioned | 6 (7.9) | 0 (0.0) | 3 (37.5) | 2 (28.6) | 0 (0.0) | 1 (33.3) | |
| Identification of search strategies for clinical questions (n = 76)[d] | 67 (88.2) | 50 (100.0) | 5 (62.5) | 2 (28.6) | 8 (100.0) | 2 (66.7) | **<0.001** |
| Identification of risk of bias assessment of the included studies (n = 76)[d] | 58 (76.3) | 50 (100.0) | 4 (50.0) | 0 (0.0) | 2 (25.0) | 2 (66.7) | **<0.001** |
| Mentioned the use of GRADE methodology | 83 (94.3) | 52 (98.1) | 12 (80.0) | 8 (100.0) | 8 (100.0) | 3 (75.0) | 0.084 |
| Identification of SoF tables | 61 (69.3) | 47 (88.7) | 5 (33.3) | 0 (0.0) | 7 (87.5) | 2 (50.0) | **<0.001** |
| The minimal important difference was established for at least one clinical question | 5 (5.7) | 5 (9.4) | 0 (0.0) | 0 (0.0) | 0 (0.0) | 0 (0.0) | 0.888 |
| EtD and criteria for resource use and cost-effectiveness | | | | | | | **<0.001** |
| Did not conduct EtD | 34 (38.6) | 12 (22.6) | 11 (73.3) | 7 (87.5) | 2 (25.0) | 2 (50.0) | |
| Conducted EtD but did not mention any local costs | 19 (21.6) | 12 (22.6) | 1 (6.7) | 0 (0.0) | 6 (75.0) | 0 (0.0) | |
| Conducted EtD and mentioned only direct costs | 31 (35.2) | 25 (47.2) | 3 (20.0) | 1 (12.5) | 0 (0.0) | 2 (50.0) | |
| Conducted EtD and specified the cost of avoiding or causing a particular outcome | 0 (0.0) | 0 (0.0) | 0 (0.0) | 0 (0.0) | 0 (0.0) | 0 (0.0) | |
| Conducted EtD and searched for economic studies | 3 (3.4) | 3 (5.7) | 0 (0.0) | 0 (0.0) | 0 (0.0) | 0 (0.0) | |
| Conducted EtD and performed economic analyses | 1 (1.1) | 1 (1.9) | 0 (0.0) | 0 (0.0) | 0 (0.0) | 0 (0.0) | |

GRADE: Grading of Recommendations, Assessment, Development, and Evaluations; EtD: Evidence to Decision; IETSI – EsSalud: Institute of Health Technology Assessment and Research of the Social Security Health Insurance; INS: National Institute of Health of the Ministry of Health; SoF: Summary of Findings.

[a]For the variables where information could not be obtained from all the CPGs, the number of CPGs from which information was available is indicated in parentheses (n=).

[b]General Directorate of Strategic Health Interventions of the Ministry of Health (DGIESP) and Peruvian Society of Neurology.

[c]Fisher's exact test.

[d]Thirteen CPGs were excluded because they were adopted (did not conduct searches or analyses to address their clinical questions).

## Discussion

Our study aimed to describe the characteristics of CPGs developed in Peru and published from 2018 to December 2023. We included 88 CPGs, which exhibited a declining trend in their number over time. Of these CPGs, 60.2% were developed by IETSI-EsSalud. We observed inter-institutional variability in the use of search engines, transparency of search strategies, and risk assessment for bias. Furthermore, 94.3% of the CPGs reported using the GRADE methodology for formulating recommendations; however, only 69.3% included SoF tables, and 61.4% utilized the EtD framework. Additionally, there was a notable lack of economic studies or evaluations within the CPGs.

We identified a downward trend in the number of published CPGs in Peru between 2018 and 2023. Part of this decline can be attributed to the COVID-19 pandemic, which could have hindered the development of meetings of CPG developer groups, previously conducted in person, and led to a redirection of health priorities (prompting decision-making organizations like IETSI-EsSalud or Ministry of Health) to produce quicker evidence synthesis documents [15,16]. Additionally, other factors could have influenced this trend, such as the instability of Peruvian public organizations [17].

Regarding methodological aspects, only 23.7% of the CPGs employed three or more search engines for evidence identification, and the same percentage did not include a bias risk assessment. These limitations are critical, as thorough search and bias assessment are essential components of SRs [18]. Their absence could potentially undermine the trustworthiness of the SRs conducted by CPGs, particularly when identifying primary studies [19].

The GRADE system provides a structured, explicit, and transparent method for evaluating evidence certainty and formulating recommendations [20]. Despite national regulations in Peru mandating that CPG development should be based on SRs and utilize the GRADE methodology for recommendations [9], significant variability exists in the CPG development process. This variability may reflect limited enforcement mechanisms and the decentralized nature of CPG development in Peru. Institutions such as EsSalud, which have internal mandates and established technical methodological teams [21], have shown greater adherence to GRADE components. In contrast, other institutions may lack technical capacity or internal regulations that reinforce adherence to these national standards. These regulatory and institutional differences likely contribute to the heterogeneity observed in the use of GRADE components across CPGs.

We found that 94.3% of the CPGs reported using the GRADE methodology for their recommendations, following regulatory standards. However, only 69.3% included SoF tables, and 61.4% performed EtD frameworks. This indicates that many Peruvian CPGs were published without fulfilling the minimum requirements outlined by the GRADE guidance group for the use of the GRADE methodology, specifically the inclusion of SoF tables and EtD frameworks [22]. This shortfall may be attributed to limited methodological capacity, particularly a lack of trained methodologists in GRADE methodology, as reported in other countries in the region, such as Brazil, where despite centralized process for CPG development, there is a persistent shortage of qualified human resources and limited investment in capacity-building for CPG development [23]. Similar capacity constraints have also been observed across Latin America. For instance, a regional analysis of CPGs published between 2011 and 2017 found that many were developed with limited technical and institutional support, insufficient training in evidence-based methodologies, and minimal application of GRADE, compromising both methodological rigor and transparency [24]. In Mexico, however, the adherence to the GRADE methodology was even lower, with only 29.1% of CPGs claiming to use it, which highlights an even greater issue in guideline implementation [5]. This issue has been similarly identified in reviews of health technology assessments in Peru (4.6%) [25]. Similar findings have also been reported in other studies examining international CPGs, such as those developed in Australia [26], China [27], and in specific areas like trauma surgery [28] and pressure injury prevention [29] that claimed to use the GRADE methodology. However, the implementation of the GRADE methodology differed across these CPGs, with some incorrectly applying GRADE and modifying certain elements of the GRADE process.

The absence of SoF tables and EtD frameworks in a substantial proportion of CPGs may reduce their practical utility for clinicians and policymakers [20]. SoF tables synthesize the key findings in an accessible format, facilitating transparent and rapid interpretation of the certainty of evidence and magnitude of effects [30]. Likewise, EtD frameworks support

decision-makers in adapting recommendations to local contexts by explicitly considering factors such as resource availability, feasibility, equity, and patient preferences [31]. Without these tools, recommendations may lack clarity, contextual relevance, or perceived credibility, potentially limiting their implementation and impact in real-world settings [31].

In addition to assessing benefits and harms, economic analysis is crucial for decision-making, especially in resource-limited settings, where judicious resource management is essential to ensure equitable decisions. The GRADE guidelines include resource use and cost-benefit considerations among the EtD criteria [32,33]. In our study, only 39.7% of CPGs assessed resource use, with only three CPGs searching for economic studies, and one conducting a budget impact analysis. Similarly, a previous study found that none of the Peruvian health technology assessments sought or conducted economic studies, and 85.5% only presented the costs of health technologies [25]. This gap may reflect structural limitations, such as a shortage of trained personnel in health economics, the absence of clear institutional methodological CPGs for economic evidence, and the lack of structured economic data from Peruvian healthcare services. These challenges are not unique to Peru and have been reported in other countries, such as Brazil, where the need to invest in capacity-building and methods development for economic evaluation in CPGs has also been emphasized [23]. Furthermore, insufficient training and fragmented institutional processes have been identified as key obstacles to CPG development across Latin America [24].

Efforts to standardize CPG development in Latin America have employed various strategies, gradually shifting toward evidence-based practices since 2014 [34]. This shift, driven by academic institutions and research groups, has prioritized guideline adaptation, adoption, or de novo development, supported by the creation of methodological manuals [34]. This approach has gained traction across the region, with countries like Brazil [35], Colombia [36], Chile [13], Mexico [5], Argentina [12], and Peru [9] establishing standards for both CPG development and implementation of CPGs [34].

In Peru, the first technical document for CPGs development, issued in 2005 by the Ministry of Health, classified evidence using the U.S. Preventive Services Task Force model but lacked robust processes for clinical question formulation or guideline adaptation [37]. For example, this document described the methodology as a literature review on the pathology of interest, applying recommendation levels based on study design and findings, without a clear and systematic process for reaching conclusions. Moreover, the adaptation of CPGs lacked a systematic approach, as no explicit criteria were established for selecting international CPGs. Additionally, the methodological process was not required to be explicitly presented; instead, the CPG document was structured as a literature summary covering aspects such as the condition's name, definition, associated risk factors, clinical presentation, diagnosis, auxiliary tests, management, complications, referral and counter-referral, flowchart or algorithm, and references [37]. The 2015 technical document marked progress by incorporating PICO (Population, Intervention, Comparator, Outcome) questions, systematic reviews, and the GRADE approach [9]. Despite improvements, challenges persist in ensuring the consistent production and implementation of evidence-based CPGs.

Based on our findings, we propose a strategic roadmap to improve the development and implementation of CPGs in Peru. Improving CPG management in Peru requires strengthening the training of methodologists in evidence synthesis, integrating economic analyses into decision-making, and ensuring greater consistency in CPG development. Addressing these methodological gaps will require coordinated efforts from key institutions such as EsSalud and the Ministry of Health of Peru. EsSalud, as the largest social health insurance provider and the institution with the most evidence-based CPGs developed using the GRADE methodology, could establish specialized teams of trained methodologists to oversee the CPG development process and ensure adherence to evidence-based standards. Meanwhile, the Ministry of Health of Peru, could implement regulatory policies mandating the use of SRs and economic evaluations in all national CPGs. Furthermore, both institutions could collaborate with academic centers to develop standardized training programs for CPG developers, promoting the use of structured methodologies and transparent decision-making processes. Another key strategy would be the creation of a centralized repository for CPGs, ensuring accessibility, periodic updates, and methodological consistency across institutions.

Our study contributes to ongoing discussions on improving CPG quality and implementation in Peru by providing a snapshot of current trends and identifying critical gaps that need to be addressed to enhance the evidence base for healthcare decision-making.

## Limitations and strengths

This study has certain limitations that should be acknowledged. Firstly, we only included CPGs that explicitly stated they had conducted SRs to formulate recommendations, aligning with current international and national definitions. As a result, documents that did not meet these criteria or were not available in full-text online were excluded from our analysis. Second, we did not use quality assessment tools such as the Appraisal of Guidelines for Research and Evaluation II (AGREE II) because our focus was on specific methodological aspects, particularly those related to GRADE, in line with the Peruvian guideline development manual, rather than an assessment of overall quality. AGREE II provides a broad evaluation framework, but its combination of multiple domains and reported limitations, such as subjective scoring and lack of weighting [38], may reduce its utility for focused methodological analysis.

Similarly, we did not evaluate the quality of recommendations, good clinical practices, conflict of interest management, adherence measurement efforts, or implementation strategies, leaving these areas for future research on Peruvian CPGs. Regarding the selection of 2018 as the starting year for assessing the CPGs, we chose this date because it is likely that the first national CPGs fully developed under the methodological standards established in 2015 [9] were published from 2018 onward. This is based on the understanding that national CPG developers began applying the GRADE methodology in 2016, allowing time for training and adaptation. However, we acknowledge that excluding older CPGs may have influenced the results of our analysis. Finally, our search strategy primarily relied on Google and publicly accessible local repositories, excluding PubMed, Scopus, and other search engines, as CPGs in Peru are not required to be published as journal articles. Consequently, some CPGs may have been omitted if they were not indexed by Google, if their identification as journal articles in the excluded search engines was faster, or if they were only accessible through institutional repositories or other less visible platforms.

Additionally, while CPGs are often grouped by health system sector (e.g., Ministry of Health, Social Security, or private sector), in this study we classified them by institution to better reflect the decentralized nature of CPG development in Peru. This approach allowed for a more accurate characterization of methodological differences among the various entities responsible for their development.

Despite these limitations, this study is the first to assess the methodological characteristics of Peruvian CPGs following the national norms published in 2015. We conducted data extraction and assessment in duplicate, ensuring reliability and standardization. Our findings shed light on methodological challenges in Peruvian CPGs, which may extend to other regions, emphasizing the need for improved oversight in CPG development and enhanced training for methodologists.

## Conclusions

In conclusion, we found a decline in the publication of CPGs in Peru, likely impacted by the COVID-19 pandemic and challenges faced by the Peruvian health systems. Many CPGs lack clear search strategies and risk of bias assessments. While many CPGs claimed adherence to the GRADE methodology, a notable percentage failed to provide SoF tables, utilize the EtD framework, or use minimal important differences; and only a few CPGs searched or performed economic analyses. Addressing these shortcomings is crucial for advancing the development of rigorous CPGs that facilitate equitable healthcare delivery in Peru.

## Supporting information

**S1 Table. Search strategies.**
(DOCX)

**S2 Table. Table of excluded CPGs.**
(DOCX)

**S3 Dataset. Dataset of Peruvian Clinical Practice Guidelines (2018–2023).**
(XLSX)

**S4 Checklist. PRISMA for Scoping Reviews (PRISMA-ScR).**
(DOCX)

## Author contributions

**Conceptualization:** Ana Brañez-Condorena, Blanca Solis-Chimoy, Alvaro Taype-Rondan.

**Data curation:** Ana Brañez-Condorena, Blanca Solis-Chimoy, Jhonatan R. Mejia, Lesly Chávez-Rimache, David R. Soriano-Moreno, Jose Ernesto Fernández-Chinguel.

**Formal analysis:** Alvaro Taype-Rondan.

**Investigation:** Ana Brañez-Condorena, Alvaro Taype-Rondan.

**Methodology:** Ana Brañez-Condorena, Blanca Solis-Chimoy, Jhonatan R. Mejia, Lesly Chávez-Rimache, David R. Soriano-Moreno, Jose Ernesto Fernández-Chinguel, Alvaro Taype-Rondan.

**Supervision:** Alvaro Taype-Rondan.

**Writing – original draft:** Ana Brañez-Condorena, Blanca Solis-Chimoy, Jhonatan R. Mejia, Lesly Chávez-Rimache, David R. Soriano-Moreno, Jose Ernesto Fernández-Chinguel, Alvaro Taype-Rondan.

**Writing – review & editing:** Ana Brañez-Condorena, Blanca Solis-Chimoy, Jhonatan R. Mejia, Lesly Chávez-Rimache, David R. Soriano-Moreno, Jose Ernesto Fernández-Chinguel, Alvaro Taype-Rondan.

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
