## [Decision Letter · Decision Letter 0]

26 Nov 2024

Dear Dr. Taype-Rondan,

Thank you for submitting your manuscript to PLOS ONE. After careful consideration, we feel that it has merit but does not fully meet PLOS ONE’s publication criteria as it currently stands. Therefore, we invite you to submit a revised version of the manuscript that addresses the points raised during the review process.

Please check the reviewers' comments and modify the manuscript accordingly. I particularly support suggestions regarding the contextualization of this issue. You should compile data about previous efforts to standardize Clinical Practice Guidelines (CPGs) in Peru and other South American countries. Regarding discussion and conclusions, what actions should be expected in the following years? How could this study contribute to improving CPG management in Peru? Please add these topics in the discussion.

We look forward to receiving your revised manuscript.

Kind regards,

Alexis G. Murillo Carrasco

Academic Editor

PLOS ONE

Journal Requirements:

I have read the journal's policy and the authors of this manuscript have the following competing interests: ABC, JRM, LCR and ATR have participated in the development of clinical practice guidelines for the Institute of Health Technology Assessment and Research (IETSI) of EsSalud and have received compensation for their work. The authors declare no additional potential conflicts of interest related to this study. 

Reviewers' comments:

Reviewer's Responses to Questions

**Comments to the Author**

1. Is the manuscript technically sound, and do the data support the conclusions?

Reviewer #1: Yes

Reviewer #2: Yes

Reviewer #3: Yes

2. Has the statistical analysis been performed appropriately and rigorously?

Reviewer #1: Yes

Reviewer #2: Yes

Reviewer #3: Yes

3. Have the authors made all data underlying the findings in their manuscript fully available?

Reviewer #1: Yes

Reviewer #2: Yes

Reviewer #3: Yes

4. Is the manuscript presented in an intelligible fashion and written in standard English?

Reviewer #1: Yes

Reviewer #2: Yes

Reviewer #3: Yes

Reviewer #1: good work done by you and your team member and also congratulation to you. .............................................................................................................................

Reviewer #2: Overall this is a good one and address some of questions I interested in. The following is my opinion that I think can make it better:

1. Please onvert all dataset variable names from Spanish to English, as shown in the provided Excel file, to improve accessibility for international researchers. This translation will help external reviewers and collaborators better understand the variables and verify the data.

2. The introduction mentions deficiencies in methodology found in previous studies. To provide context, consider creating a table or a brief review summarizing these past methodological shortcomings. Then, compare these with the findings from your current study. This addition would be beneficial for future researchers by highlighting patterns and supporting efforts to standardize Clinical Practice Guidelines (CPGs) in South America.

3. What are the standards for CPGs in Peru versus other countries? Provide an overview of international standards for CPGs, if available, and discuss the differences with Peruvian standards. Consider sharing your perspective on what constitutes an ideal standard for CPGs, with a critical analysis. For instance, how might the CPGs from countries like Japan or the U.S. be considered superior, and what aspects could be improved?

4. Expand on the data collection process, specifically the consensus method. Include details on how consensus was reached, the criteria used, and the process undertaken to ensure quality in data collection. This additional information would help demonstrate the rigor applied to maintain data reliability.

5. In Table 3, in addition to the p-values and percentages, please include the standard deviation and 95% confidence intervals for each result. This added statistical detail will provide a clearer picture of the variability and confidence in the results.

Reviewer #3: This is an interesting scoping review that describes the methodological characteristics of Peruvian clinical practice guidelines. To further strengthen the manuscript, please see the comments and suggestions below.

Introduction

• The Introduction lacks literature on previous studies.

• Additional background and explanation on what is considered as an ideal/standard CPG and what is the standard content of CPGs would be helpful to provide additional context. Also, it would be helpful if the authors specify what methodological deficiencies are usually observed in local CPGs.

Methods

• Page 6, Line 80: Any rationale for considering 2018 as the start year for assessing the CPGs when the standardized methodology document from Ministry of Health was established in 2015?

• Was the literature search not conducted in other search engines including PubMed, SCOPUS etc.?

• Why were CPGs with partial updates not included? Wouldn’t the partial updates also indicate changes based on the 2015 standardized methodology from Ministry of Health?

Results

• Any reason the sample size (n) was equal to 85 and not 88 for Tables 2 and 3?

Discussion

• The Discussion should highlight the implications of this study and how the results would help in bringing positive changes related to CPGs.

**Do you want your identity to be public for this peer review?** For information about this choice, including consent withdrawal, please see our Privacy Policy

Reviewer #1: No

Reviewer #2: No

Reviewer #3: No

---

## [Author Response · Author response to Decision Letter 1]

10 Jan 2025

Editor’s comments:

1. You should compile data about previous efforts to standardize Clinical Practice Guidelines (CPGs) in Peru and other South American countries. Regarding discussion and conclusions, what actions should be expected in the following years? How could this study contribute to improving CPG management in Peru? Please add these topics in the discussion.

Answer: Thanks for your comment. We have added the following information: “Efforts to standardize CPG development in Latin America have employed various strategies, gradually shifting toward evidence-based practices since 2014 [30]. This shift, driven by academic institutions and research groups, has prioritized guideline adaptation, adoption, or de novo development, supported by the creation of methodological manuals [30]. This approach has gained traction across the region, with countries like Brazil [31], Colombia [32], Chile [13], Mexico [5], Argentina [12], and Peru [9] establishing standards for both CPG development and implementation of CPGs [30]. In Peru, the first technical document, issued in 2005, classified evidence using the U.S. Preventive Services Task Force model but lacked robust processes for clinical question formulation or guideline adaptation [33]. The 2015 technical document marked progress by incorporating PICO (Population, Intervention, Comparator, Outcome) questions, systematic reviews, and the GRADE approach [9]. Despite improvements, challenges persist in ensuring the consistent production and implementation of evidence-based CPGs.

Improving CPG management in Peru should focus on enhancing the training of methodologists in evidence synthesis, increasing the integration of economic analysis in decision-making, and ensuring greater consistency in guideline development processes. In the coming years, expected actions include greater collaboration between institutions, strengthening oversight mechanisms, and the continued development of robust training programs for CPG developers. Our study contributes to ongoing discussions on improving CPG quality and implementation in Peru by providing a snapshot of current trends and identifying critical gaps that need to be addressed to enhance the evidence base for healthcare decision-making.”

Journal Requirements:

Answer: We verified that our manuscript follows PLOS ONE’s style requirements.

I have read the journal's policy and the authors of this manuscript have the following competing interests: ABC, JRM, LCR and ATR have participated in the development of clinical practice guidelines for the Institute of Health Technology Assessment and Research (IETSI) of EsSalud and have received compensation for their work. The authors declare no additional potential conflicts of interest related to this study.

Answer: We have updated the competing interests section and included it in the cover letter: “ABC, JRM, LCR and ATR have participated in the development of clinical practice guidelines for the Institute of Health Technology Assessment and Research (IETSI) of EsSalud and have received compensation for their work. The authors declare no additional potential conflicts of interest related to this study. This does not alter our adherence to PLOS ONE policies on sharing data and materials”.

Reviewers’ comments:

R2.1: Please convert all dataset variable names from Spanish to English, as shown in the provided Excel file, to improve accessibility for international researchers. This translation will help external reviewers and collaborators better understand the variables and verify the data.

Answer: We changed all dataset variable names to English.

R2.2: The introduction mentions deficiencies in methodology found in previous studies. To provide context, consider creating a table or a brief review summarizing these past methodological shortcomings. Then, compare these with the findings from your current study. This addition would be beneficial for future researchers by highlighting patterns and supporting efforts to standardize Clinical Practice Guidelines (CPGs) in South America.

Answer: Thanks for your comment. We have added the following information to the Introduction section: “Common shortcomings include unclear processes for formulating recommendations [4-6], inadequate systematic searches and selection criteria [4,5], limited consideration of evidence strengths and limitations [4-6], and failure to incorporate patients' preferences [4,6]. In Peru, an evaluation of 17 CPGs published by the Ministry of Health between 2009 and 2014 revealed similar deficiencies [7], including a lack of information about the guideline development team, minimal use of evidence, and unclear recommendation processes [7]. A further study of 31 CPGs developed by the Peruvian Social Health Insurance System (EsSalud) up to December 2015 reported gaps in evidence-based frameworks and inconsistent reporting of recommendations [8].”

Also, we have added the following information to the Discussion section: “This shortfall may be attributed to a lack of trained methodologists in GRADE methodology, as reported in other countries in the region, such as Brazil [22], In Mexico, however, the adherence to the GRADE methodology was even lower, with only 29.1% of CPGs claiming to use it, which highlights an even greater issue in guideline implementation [5]. This issue has been similarly identified in reviews of health technology assessments in Peru (4.6%) [23].”

R2.3: What are the standards for CPGs in Peru versus other countries? Provide an overview of international standards for CPGs, if available, and discuss the differences with Peruvian standards. Consider sharing your perspective on what constitutes an ideal standard for CPGs, with a critical analysis. For instance, how might the CPGs from countries like Japan or the U.S. be considered superior, and what aspects could be improved?

Answer: Thanks for your comment. We have added the following information in the Introduction section: “Internationally, institutions such as the World Health Organization (WHO) [10], National Institute for Health and Care Excellence (NICE) in the UK [11], and regional organizations such as the Ministry of Health of Argentina [12] and Chile [13] follow rigorous methodologies emphasizing transparency, evidence quality assessment, and regular updates. While Peru's standards have advanced, they could benefit from greater alignment with these international practices, ensuring guidelines remain evidence-based and relevant.”

We also have added information in the Discussion section: “Efforts to standardize CPG development in Latin America have employed various strategies, gradually shifting toward evidence-based practices since 2014 [30]. This shift, driven by academic institutions and research groups, has prioritized guideline adaptation, adoption, or de novo development, supported by the creation of methodological manuals [30]. This approach has gained traction across the region, with countries like Brazil [31], Colombia [32], Chile [13], Mexico [5], Argentina [12], and Peru [9] establishing standards for both CPG development and implementation of CPGs [30]. In Peru, the first technical document, issued in 2005, classified evidence using the U.S. Preventive Services Task Force model but lacked robust processes for clinical question formulation or guideline adaptation [33]. The 2015 technical document marked progress by incorporating PICO (Population, Intervention, Comparator, Outcome) questions, systematic reviews, and the GRADE approach [9]. Despite improvements, challenges persist in ensuring the consistent production and implementation of evidence-based CPGs.

Improving CPG management in Peru requires a multi-faceted approach, including increased investment in training programs for methodologists, the creation of a central oversight body for quality assurance, and collaboration with international partners to adopt best practices. Future efforts should also focus on expanding access to data for economic evaluations and developing tailored tools to facilitate systematic resource use assessments.”

R2.4: Expand on the data collection process, specifically the consensus method. Include details on how consensus was reached, the criteria used, and the process undertaken to ensure quality in data collection. This additional information would help demonstrate the rigor applied to maintain data reliability.

Answer: Thanks for your comment. We added the following information: “In case of disagreements, consensus was reached through discussion among the investigators, followed by a second review of the data from the included CPGs to identify the precise information. To ensure the quality of the data collection process, a predefined set of criteria was applied to each variable. This included cross-checking the extracted information with the original documents, verifying consistency across multiple versions and journal articles of the CPGs, and ensuring the accuracy of all recorded data. If consensus could not be reached, another researcher (ATR) was consulted to resolve any uncertainties or clarify discrepancies in the data.”

R2.5: In Table 3, in addition to the p-values and percentages, please include the standard deviation and 95% confidence intervals for each result. This added statistical detail will provide a clearer picture of the variability and confidence in the results.

Answer: Thank you for your suggestion. However, the variables presented in Table 3 are categorical and expressed as percentages. Therefore, it is not possible to calculate standard deviations or 95% confidence intervals for these results.

R3.1: Introduction

• The Introduction lacks literature on previous studies.

Answer: We detailed the information of previous studies: “]. In Peru, an evaluation of 17 CPGs published by the Ministry of Health between 2009 and 2014 revealed similar deficiencies [7], including a lack of information about the guideline development team, minimal use of evidence, and unclear recommendation processes [7]. A further study of 31 CPGs developed by the Peruvian Social Health Insurance System (EsSalud) up to December 2015 reported gaps in evidence-based frameworks and inconsistent reporting of recommendations [8].”

R3.2: Introduction

• Additional background and explanation on what is considered as an ideal/standard CPG and what is the standard content of CPGs would be helpful to provide additional context. Also, it would be helpful if the authors specify what methodological deficiencies are usually observed in local CPGs.

Answer: Thanks for your comment. We added the following information: “A trustworthy CPG adheres to these standards that emphasize key components including clearly defined clinical questions, systematic literature searches, the use of a grading system for evidence, explicit criteria for formulating recommendations, external review, and a structured process for updates [1,2].”

R3.3: Methods

• Page 6, Line 80: Any rationale for considering 2018 as the start year for assessing the CPGs when the standardized methodology document from Ministry of Health was established in 2015?

Answer: Thanks for your comment. We considered it reasonable that, although the new standards were established in 2015, they would begin to be applied starting in 2016, allowing time for training and adaptation. Therefore, the first guidelines fully developed under these standards would likely be published from 2018 onward, making this a more appropriate starting point for our analysis.

R3.4: Methods

• Was the literature search not conducted in other search engines including PubMed, SCOPUS etc.?

Answer: Thanks for your comment. In Peru, clinical practice guidelines are not required to be published as journal articles. When they are published in this format, they often present only a summary (a brief justification for each recommendation) of the guideline content rather that the complete document. Therefore, we prioritized searches on local websites, Google, and Google Scholar, where full versions of the guidelines are more likely to be available. This approach allowed us to access the comprehensive documents necessary for our analysis.

R3.5: Methods

• Why were CPGs with partial updates not included? Wouldn’t the partial updates also indicate changes based on the 2015 standardized methodology from Ministry of Health?

Answer: Thanks for your comment. Since the standardized methodology was predominantly implemented from 2018 onward, there have not been frequent updates to previous guidelines published under this framework. We identified only two partially updated guidelines developed by the Social Health Insurance of Peru (IETSI-EsSalud). which are listed in the “S2 Table: Table of excluded CPGs”. The first versions of these guidelines, published in 2018, already followed the GRADE methodology, aligning with the standardized approach. The partial updates focused primarily on updating evidence supporting specific recommendations that could influence their direction (e.g., recommend/suggest or not recommend/suggest), rather than revising the overall methodology. Therefore, we deemed it unnecessary to include these partial updates, as they did not represent comprehensive methodological changes.

R3.6: Results

• Any reason the sample size (n) was equal to 85 and not 88 for Tables 2 and 3?

Answer: Thank you for your comment. The sample size (n) in Table 2 and Table 3 is 88, not 85. We have made the necessary corrections. Additionally, we re-verified all the information in the Tables regarding percentage calculations (%), and only two corrections were made in Table 1 and Table 2.

R3.7: Discussion

• The Discussion should highlight the implications of this study and how the results would help in bringing positive changes related to CPGs.

Answer: We have added the following information: “Efforts to standardize CPG development in Latin America have employed various strategies, gradually shifting toward evidence-based practices since 2014 [30]. This shift, driven by academic institutions and research groups, has prioritized guideline adaptation, adoption, or de novo development, supported by the creation of methodological manuals [30]. This approach has gained traction across the region, with countries like Brazil [31], Colombia [32], Chile [13], Mexico [5], Argentina [12], and Peru [9] establishing standards for both CPG development and implementation of CPGs [30]. In Peru, the first technical document, issued in 2005, classified evidence using the U.S. Preventive Services Task Force model but lacked robust processes for clinical question formulation or guideline adaptation [33]. The 2015 technical document marked progress by incorporating PICO (Population, Intervention, Comparator, Outcome) questions, systematic reviews, and the GRADE approach [9]. Despite improvements, challenges persist in ensuring the consistent production and implementation of evidence-based CPGs.

Improving CPG management in Peru should focus on enhancing the training of methodologists in evidence synthesis, increasing the integration of economic analysis in decision-making, and ensuring greater consistency in guideline development processes

---

## [Decision Letter · Decision Letter 1]

27 Jan 2025

Dear Dr. Taype-Rondan,

Thank you for submitting your manuscript to PLOS ONE. After careful consideration, we feel that it has merit but does not fully meet PLOS ONE’s publication criteria as it currently stands. Therefore, we invite you to submit a revised version of the manuscript that addresses the points raised during the review process.

In addition to the reviewer response already provided, please state in the limitations of this study your rationale for considering 2018 as the start year for assessing the CPGs and for not including Pubmed, Scopus, or other search engines. 

Please discuss more on suggestions to adress methodological issues in Peru, especially on how ESSALUD or Peruvian Health Minister could contribute to improve the elaboration of CPGs.

Provide examples illustrating how the Peruvian first technical document lacked robust processes for formulating clinical questions or adapting guidelines. Please review the entire document ensuring all authors' comments are objective and highlight the methods used to identify shortcomings, specifying particular flaws where applicable.

We look forward to receiving your revised manuscript.

Kind regards,

Alexis G. Murillo Carrasco

Academic Editor

PLOS ONE

Journal Requirements:

Reviewers' comments:

Reviewer's Responses to Questions

**Comments to the Author**

Reviewer #2: All comments have been addressed

Reviewer #3: All comments have been addressed

2. Is the manuscript technically sound, and do the data support the conclusions?

Reviewer #2: Yes

Reviewer #3: Yes

3. Has the statistical analysis been performed appropriately and rigorously?

Reviewer #2: Yes

Reviewer #3: Yes

4. Have the authors made all data underlying the findings in their manuscript fully available?

Reviewer #2: Yes

Reviewer #3: Yes

5. Is the manuscript presented in an intelligible fashion and written in standard English?

Reviewer #2: Yes

Reviewer #3: Yes

Reviewer #2: This resubmission effectively addresses all of my and the other reviewers' previous concerns. The authors have demonstrated a thorough and systematic effort to resolve the issues raised, and the revised manuscript now provides a comprehensive overview of Peruvian CPG development and related methodological characteristics. I appreciate the detailed responses and thoughtful additions to the manuscript.

However, upon further review, I believe the manuscript could be strengthened by explicitly acknowledging a potential bias in the search strategy. Specifically, as stated in the response: "In Peru, clinical practice guidelines are not required to be published as journal articles. When they are published in this format, they often present only a summary (a brief justification for each recommendation) of the guideline content rather than the complete document." This raises the concern that some relevant documents may not have been retrieved using Google as the primary search platform. There is a risk of missing guidelines that are either not indexed by Google or require access through less visible platforms or institutional repositories.

I recommend adding a discussion point in the limitations section to explicitly address this potential bias and the limitations of the current search methodology. Furthermore, considering more comprehensive and systematic search techniques—such as web scraping or leveraging specialized databases—could be a valuable direction for future research. This approach would enhance the robustness of guideline retrieval and ensure better coverage.

Overall, the authors have done an excellent job with this scoping review, and the manuscript offers valuable insights into Peruvian CPG development. I am confident in recommending it for acceptance with the above minor addition.

Reviewer #3: (No Response)

**Do you want your identity to be public for this peer review?** For information about this choice, including consent withdrawal, please see our Privacy Policy

Reviewer #2: No

Reviewer #3: No

---

## [Author Response · Author response to Decision Letter 2]

10 Feb 2025

Editor’s comments:

1. In addition to the reviewer response already provided, please state in the limitations of this study your rationale for considering 2018 as the start year for assessing the CPGs and for not including Pubmed, Scopus, or other search engines.

Answer: Thanks for your comment. We added the following information in the Discussion section: “Regarding the selection of 2018 as the starting year for assessing the CPGs, we chose this date because it is likely that the first national CPGs fully developed under the methodological standards established in 2015 (9) were published from 2018 onward. This is based on the understanding that national CPG developers began applying the GRADE methodology in 2016, allowing time for training and adaptation. However, we acknowledge that excluding older CPGs may have influenced the results of our analysis.

Finally, our search strategy primarily relied on Google and publicly accessible local repositories, excluding PubMed, Scopus, and other search engines, as CPGs in Peru are not required to be published as journal articles. Consequently, some CPGs may have been omitted if they were not indexed by Google, if their identification as journal articles in the excluded search engines was faster, or if they were only accessible through institutional repositories or other less visible platforms.”

2. Please discuss more on suggestions to adress methodological issues in Peru, especially on how ESSALUD or Peruvian Health Minister could contribute to improve the elaboration of CPGs.

Answer: Thanks for your comment. We added the following information: “Improving CPG management in Peru requires strengthening the training of methodologists in evidence synthesis, integrating economic analyses into decision-making, and ensuring greater consistency in CPG development. Addressing these methodological gaps will require coordinated efforts from key institutions such as EsSalud and the Ministry of Health of Peru. EsSalud, as the largest social health insurance provider and the institution with the most evidence-based CPGs developed using the GRADE methodology, could establish specialized teams of trained methodologists to oversee the CPG development process and ensure adherence to evidence-based standards. Meanwhile, the Ministry of Health of Peru, could implement regulatory policies mandating the use of SRs and economic evaluations in all national CPGs. Furthermore, both institutions could collaborate with academic centers to develop standardized training programs for CPG developers, promoting the use of structured methodologies and transparent decision-making processes. Another key strategy would be the creation of a centralized repository for CPGs, ensuring accessibility, periodic updates, and methodological consistency across institutions.”

3. Provide examples illustrating how the Peruvian first technical document lacked robust processes for formulating clinical questions or adapting guidelines. Please review the entire document ensuring all authors' comments are objective and highlight the methods used to identify shortcomings, specifying particular flaws where applicable.

Answer: Thanks for your comment. We added the following information: “For example, this document described the methodology as a literature review on the pathology of interest, applying recommendation levels based on study design and findings, without a clear and systematic process for reaching conclusions. Moreover, the adaptation of CPGs lacked a systematic approach, as no explicit criteria were established for selecting international CPGs. Additionally, the methodological process was not required to be explicitly presented; instead, the CPG document was structured as a literature summary covering aspects such as the condition's name, definition, associated risk factors, clinical presentation, diagnosis, auxiliary tests, management, complications, referral and counter-referral, flowchart or algorithm, and references [33].”

Journal Requirements:

Answer: We have verified all citations and found no references to retracted papers.

Reviewers’ comments:

R2.1: However, upon further review, I believe the manuscript could be strengthened by explicitly acknowledging a potential bias in the search strategy. Specifically, as stated in the response: "In Peru, clinical practice guidelines are not required to be published as journal articles. When they are published in this format, they often present only a summary (a brief justification for each recommendation) of the guideline content rather than the complete document." This raises the concern that some relevant documents may not have been retrieved using Google as the primary search platform. There is a risk of missing guidelines that are either not indexed by Google or require access through less visible platforms or institutional repositories.

I recommend adding a discussion point in the limitations section to explicitly address this potential bias and the limitations of the current search methodology. Furthermore, considering more comprehensive and systematic search techniques—such as web scraping or leveraging specialized databases—could be a valuable direction for future research. This approach would enhance the robustness of guideline retrieval and ensure better coverage.

Answer: Thanks for your comment. We added the following information in Discussion section: “Finally, our search strategy primarily relied on Google and publicly accessible local repositories, excluding PubMed, Scopus, and other search engines, as CPGs in Peru are not required to be published as journal articles. Consequently, some CPGs may have been omitted if they were not indexed by Google, if their identification as journal articles in the excluded search engines was faster, or if they were only accessible through institutional repositories or other less visible platforms.”

---

## [Decision Letter · Decision Letter 2]

30 Apr 2025

Dear Dr. Taype-Rondan,

Thank you for submitting your manuscript to PLOS ONE. After careful consideration, we feel that it has merit but does not fully meet PLOS ONE’s publication criteria as it currently stands. Therefore, we invite you to submit a revised version of the manuscript that addresses the points raised during the review process.

**ACADEMIC EDITOR:**

Kind regards,

César Félix Cayo-Rojas, Ph.D.

Academic Editor

PLOS ONE

Reviewers' comments:

Reviewer's Responses to Questions

**Comments to the Author**

Reviewer #2: All comments have been addressed

Reviewer #4: (No Response)

Reviewer #5: (No Response)

2. Is the manuscript technically sound, and do the data support the conclusions?

Reviewer #2: Yes

Reviewer #4: Partly

Reviewer #5: Yes

3. Has the statistical analysis been performed appropriately and rigorously?

Reviewer #2: Yes

Reviewer #4: No

Reviewer #5: Yes

4. Have the authors made all data underlying the findings in their manuscript fully available?

Reviewer #2: Yes

Reviewer #4: Yes

Reviewer #5: Yes

5. Is the manuscript presented in an intelligible fashion and written in standard English?

Reviewer #2: Yes

Reviewer #4: Yes

Reviewer #5: Yes

Reviewer #2: The manuscript is solid and good. I recommend this to the journal. I believed this topic will give some new perspective to the field.

Reviewer #4: 1. Potential selection bias toward tertiary-level guidelines

The study appears to be skewed toward clinical practice guidelines (CPGs) developed for tertiary-level care. It is unclear whether this is a reflection of availability or if there was an intentional search strategy targeting this level. The authors should clarify whether their inclusion criteria or sources were likely to miss guidelines developed for primary or secondary care.

2. Institution-based classification vs. Health sector grouping

The decision to classify guidelines by specific institutions (e.g., IETSI, AUNA) may fragment the analysis and limit broader policy implications. Grouping by major health system sectors—such as social security, Ministry of Health, or private sector—could provide a more comprehensive and policy-relevant perspective.

3. Absence of a critical appraisal of methodological quality beyond grade

While the focus is on GRADE components, the lack of a broader critical appraisal (e.g., AGREE II domains) limits the understanding of the overall methodological rigor of the guidelines. Including or at least discussing this could offer a more balanced assessment.

4. Lack of contextual data on developers' capacities

The authors attribute deficiencies to limited methodological capacity but do not provide data to support this claim. Information on the availability of trained personnel, institutional mandates, or access to GRADE training would strengthen the interpretation.

5. Need for exploration of policy or regulatory drivers

The manuscript could benefit from examining whether national regulations or incentive structures influence the adoption of GRADE components. This might help explain heterogeneity between institutions or sectors.

6. Unclear justification for excluding non-grade guidelines

Excluding guidelines that did not claim to use GRADE may overlook potentially methodologically sound documents. The authors should clarify the rationale and implications of this exclusion on the comprehensiveness of the review.

7. Limited Exploration of Guideline Topics and Their Alignment with National burden of disease

The paper does not analyze whether the guidelines reviewed align with national health priorities or the burden of disease in Peru. This would help assess their relevance and strategic focus.

8. Narrative synthesis could be strengthened with quantitative visualization

A table or heatmap comparing GRADE components across institutions would enhance the clarity and accessibility of the findings, particularly for policymakers and non-specialist readers.

9. Insufficient discussion of implications for end-user utility

While the absence of Summary of Findings (SoF) tables and EtD frameworks is highlighted, the practical implications for clinicians or decision-makers are not fully elaborated. This connection should be strengthened.

10. Need for a forward-looking roadmap or recommendations

The discussion section would benefit from a more detailed set of recommendations or a roadmap for improving CPG development capacity in Peru, including possible institutional partnerships, technical training, or policy reforms.

Reviewer #5: Results

The table format should follow the journal's guidelines. It is recommended to adhere to the journal’s specific instructions or protocols for creating descriptive tables.

In row 142, it is not necessary to explain what CPG stands for, as it has already been defined in the introduction and abstract.

Please verify that all percentages add up to 100%. For example, in Table 1 (“Local organizations…”, “Area addressed”), the total is 100.1%.

In Table 3, for the first variable (“Search engines used”), data from 76 CPGs is used, and 13 are marked as excluded. This presents an inconsistency with the stated total number of included CPGs (n = 88).

Discussion

Why are the results related to methodological differences in the “Identification of SoF tables” and “EtD and criteria for resource use and cost-effectiveness” not presented or discussed? (This also applies to the results section.)

Although the focus is on CPGs from Peru, it is recommended to broaden the scope or highlight the relevance of the findings so that they may serve as a reference for other countries.

Additionally, consider including as a limitation the fact that only CPGs with publicly available full-text versions were included.

In Figure 1, please indicate the main reasons why the 206 documents were excluded.

In Figure 2, include the percentage values for each color segment of the columns. This will make the figure easier to interpret. It is also important to show the sample size (“n”) for each year, as the results are currently presented in grouped form.

**Do you want your identity to be public for this peer review?** For information about this choice, including consent withdrawal, please see our Privacy Policy

Reviewer #2: No

Reviewer #4: No

Reviewer #5: No

---

## [Author Response · Author response to Decision Letter 3]

9 Jul 2025

Response letter

Dear Editor and reviewers,

Thank you for your kind consideration. In this letter, we proceed to respond to the reviewers’ comments.

Sincerely,

Alvaro Taype-Rondan, corresponding author

Reviewers’ comments:

R4.1: Potential selection bias toward tertiary-level guidelines

The study appears to be skewed toward clinical practice guidelines (CPGs) developed for tertiary-level care. It is unclear whether this is a reflection of availability or if there was an intentional search strategy targeting this level. The authors should clarify whether their inclusion criteria or sources were likely to miss guidelines developed for primary or secondary care.

Answer: We added the following clarification in the Eligibility criteria section: “No restriction was applied regarding the level of healthcare (primary, secondary, or tertiary) for which the CPGs were intended.” This addition clarifies that our inclusion criteria and search strategy were not designed to target or exclude guidelines based on the level of care.

R4.2: Institution-based classification vs. Health sector grouping

The decision to classify guidelines by specific institutions (e.g., IETSI, AUNA) may fragment the analysis and limit broader policy implications. Grouping by major health system sectors—such as social security, Ministry of Health, or private sector—could provide a more comprehensive and policy-relevant perspective.

Answer: Thank you for your observation. We agree that grouping by major health system sectors can be useful for policy analysis. However, in Peru, the development of clinical practice guidelines is highly decentralized. Each institution, whether public or private, operates independently and follows its own processes for guideline development.

For example, IETSI is solely responsible for CPG development within EsSalud (the national social security system). In contrast, the Ministry of Health has at least three separate entities involved: DGIESP (part of the Ministry’s central office), INS (its research branch), and individual hospitals, which vary in methodological capacity. Similarly, private organizations such as AUNA and the Peruvian Society of Neurology develop CPGs through their own independent teams.

Given this institutional heterogeneity, we chose to classify CPGs by developing institutions rather than by sector. This approach more accurately reflects current practices and allows identification of methodological strengths and gaps at the institutional level, differences that might be masked by broader sector-level groupings.

We added the following clarifications to the manuscript:

• At the Introduction section: “Despite the existence of this standard, CPG development in Peru remains decentralized. Different institutions, even within the same sector, develop CPGs independently, with distinct methodological teams and varying capacities. This institutional heterogeneity can influence the methodological quality and consistency of CPGs across the country”.

• At the Data extraction section: “Variables of interest included the year of publication, covered area, developing organization (given that Peruvian health institutions operate with distinct processes and capacities)...” and “CPGs were classified according to the developing institution, rather than by broader health system sectors, to more accurately reflect the decentralized structure and variability of CPG development in Peru.”

• At the Limitations section: “Additionally, while CPGs are often grouped by health system sector (e.g., Ministry of Health, Social Security, or private sector), in this study we classified them by institution to better reflect the decentralized nature of CPG development in Peru. This approach allowed for a more accurate characterization of methodological differences among the various entities responsible for their development.”

R4.3: Absence of a critical appraisal of methodological quality beyond grade

While the focus is on GRADE components, the lack of a broader critical appraisal (e.g., AGREE II domains) limits the understanding of the overall methodological rigor of the guidelines. Including or at least discussing this could offer a more balanced assessment.

Answer: Thank you for your suggestion. While we acknowledge the value of AGREE II as a comprehensive tool for appraising guideline quality, our study focused on specific methodological characteristics, particularly those related to the use of GRADE. This focus aligns with the Peruvian guideline development manual, which explicitly recommends using GRADE to assess the certainty of evidence and formulate recommendations.

We believe that AGREE II, by combining multiple domains, including reporting quality and editorial Independence, may dilute the focus on core methodological aspects. Our intention was not to provide a broad appraisal, but rather to identify and describe concrete methodological shortcomings that may compromise the reliability of the recommendations.

In addition, as highlighted in a recent systematic review [38], AGREE II presents several limitations, such as the lack of guidance for the two overall assessments, the subjectivity in scoring, and the equal weighting of all items regardless of their relevance. These issues reduce its precision for detailed methodological evaluations. We have clarified this rationale in the limitations section of the manuscript: “Second, we did not use quality assessment tools such as AGREE II because our focus was on specific methodological aspects, particularly those related to GRADE, in line with the Peruvian guideline development manual, rather than an assessment of overall quality. AGREE II provides a broad evaluation framework, but its combination of multiple domains and reported limitations, such as subjective scoring and lack of weighting [38], may reduce its utility for focused methodological analysis.”

38. Hoffmann-Eßer W, Siering U, Neugebauer EA, Brockhaus AC, Lampert U, Eikermann M. Guideline appraisal with AGREE II: Systematic review of the current evidence on how users handle the 2 overall assessments. PLoS One. 2017;12(3):e0174831. doi: 10.1371/journal.pone.0174831.

R4.4: Lack of contextual data on developers' capacities

The authors attribute deficiencies to limited methodological capacity but do not provide data to support this claim. Information on the availability of trained personnel, institutional mandates, or access to GRADE training would strengthen the interpretation.

Answer: In the manuscript, we now discuss how limited methodological capacity, including insufficient training in GRADE, has been identified as a major challenge in guideline development in other countries.

We have clarified this in the discussion section: “This shortfall may be attributed to limited methodological capacity, particularly a lack of trained methodologists in GRADE methodology, as reported in other countries in the region, such as Brazil, where despite centralized process for CPG development, there is a persistent shortage of qualified human resources and limited investment in capacity-building for CPG development [23]. Similar capacity constraints have also been observed across Latin America. For instance, a regional analysis of CPGs published between 2011 and 2017 found that many were developed with limited technical and institutional support, insufficient training in evidence-based methodologies, and minimal application of GRADE, compromising both methodological rigor and transparency [24].”

Also, we added the following information: “This gap may reflect structural limitations, such as a shortage of trained personnel in health economics, the absence of clear institutional methodological CPGs for economic evidence, and the lack of structured economic data from Peruvian healthcare services. These challenges are not unique to Peru and have been reported in other countries, such as Brazil, where the need to invest in capacity-building and methods development for economic evaluation in CPGs has also been emphasized [22] Furthermore, insufficient training and fragmented institutional processes have been identified as key obstacles to CPG development across Latin America [24].”

23. Colpani V, Kowalski SC, Stein AT, Buehler AM, Zanetti D, Côrtes G, et al. Clinical practice guidelines in Brazil – developing a national programme. Health Res Policy Sys. 2020;18(1):69. https://doi.org/10.1186/s12961-020-00582-0

24. Cabrera PA, Pardo R. Review of evidence based clinical practice guidelines developed in Latin America and Caribbean during the last decade: an analysis of the methods for grading quality of evidence and topic prioritization. Global Health. 2019;15(1):14. doi: 10.1186/s12992-019-0455-0.

R4.5: Need for exploration of policy or regulatory drivers

The manuscript could benefit from examining whether national regulations or incentive structures influence the adoption of GRADE components. This might help explain heterogeneity between institutions or sectors.

Answer: We have now added a paragraph in the Discussion section: “The GRADE system provides a structured, explicit, and transparent method for evaluating evidence certainty and formulating recommendations [20]. Despite national regulations in Peru mandating that CPG development should be based on SRs and utilize the GRADE methodology for recommendations [9], significant variability exists in the CPG development process. This variability may reflect limited enforcement mechanisms and the decentralized nature of CPG development in Peru. Institutions such as EsSalud, which have internal mandates and established technical methodological teams [21], have shown greater adherence to GRADE components. In contrast, other institutions may lack technical capacity or internal regulations that reinforce adherence to these national standards. These regulatory and institutional differences likely contribute to the heterogeneity observed in the use of GRADE components across CPGs.”

9. Ministerio de Salud de Perú. Documento técnico: Metodología para la elaboración de guías de práctica clínica. Lima: Ministerio de Salud de Perú; 2015 [cited 2025 Jan 10]. Available from: https://www.gob.pe/institucion/minsa/informes-publicaciones/314118-documento-tecnico-metodologia-para-la-elaboracion-de-guias-de-practica-clinica

20. Guyatt G, Oxman AD, Akl EA, Kunz R, Vist G, Brozek J, et al. GRADE guidelines: 1. Introduction—GRADE evidence profiles and summary of findings tables. J Clin Epidemiol. 2011;64(4):383–94. https://doi.org/10.1016/j.jclinepi.2010.04.026

21. Instituto de Evaluación de Tecnologías en Salud e Investigación (IETSI). Resolution No. 063-IETSI-ESSALUD-2021: Instruction No. 001-IETSI-ESSALUD-2021 for the development of clinical practice guidelines in EsSalud [Internet]. Lima: EsSalud; 2021 [cited 2025 Jul 5]. Available from: https://www.essalud.gob.pe/ietsi/pdfs/tecnologias_sanitarias/RS_063_IETSI_2021.pdf

R4.6: Unclear justification for excluding non-grade guidelines

Excluding guidelines that did not claim to use GRADE may overlook potentially methodologically sound documents. The authors should clarify the rationale and implications of this exclusion on the comprehensiveness of the review.

Answer: We would like to clarify that our inclusion criteria did not require CPGs to explicitly state that they used GRADE. Instead, we included CPGs that reported basing their recommendations on systematic reviews (SRs), as required by Peruvian regulations since 2015 [9].

Our objective was to assess whether CPGs that claimed to use evidence-based methods (i.e., based their recommendations on SRs) incorporated core methodological components aligned with GRADE standards. This approach allowed us to evaluate the rigor and transparency of methodological implementation among CPGs that expressed an evidence-based intent.

We have clarified this rationale in the Eligibility criteria section: “We included Peruvian CPGs that reported basing their recommendations on SRs, an essential component of evidence-based CPG development. This approach allowed us to focus on CPGs with an evidence-based intent and assess whether they incorporated key methodological components aligned with international standards [10-13] and the national regulations for CPG development in Peru [9].”

In addition, we evaluated whether each CPG explicitly mentioned the use of the GRADE methodology (Table 3, 4th variable). Although most CPGs did so (94.3%), a small number of included CPGs based their recommendations on SRs without referring to GRADE. Therefore, our inclusion criterion was not restricted to the use of GRADE, but rather to the presence of SR-based recommendations, which allowed us to explore both the adoption and the use of GRADE components in eligible CPGs.

R4.7: Limited Exploration of Guideline Topics and Their Alignment with National burden of disease

The paper does not analyze whether the guidelines reviewed align with national health priorities or the burden of disease in Peru. This would help assess their relevance and strategic focus.

Answer: Thanks for your comment. We agree that analyzing the alignment of CPG topics with national health priorities and disease burden is an important area of exploration. However, this was beyond the scope of our study, which focused specifically on assessing methodological characteristics of CPGs, particularly regarding their use of evidence-based components such as SRs and GRADE methodology. Future research could explore the concordance between the content of Peruvian CPGs and the country's epidemiological profile, which would provide valuable insights into the strategic focus and relevance of national guideline development.

R4.8: Narrative synthesis could be strengthened with quantitative visualization

A table or heatmap comparing GRADE components across institutions would enhance the clarity and accessibility of the findings, particularly for policymakers and non-specialist readers.

Answer: Thank you for your suggestion. We agree that visual tools can help synthesize complex data. However, we opted to present our findings in detailed percentage-based tables (Tables 2 and 3), which we believe provide more precise and interpretable information, especially for readers who require exact data to inform policy or methodological decisions. A heatmap may obscure the magnitude of differences, particularly for components with low or moderate frequencies. Given the diversity of methodological practices across institutions, we considered that a tabular format would better support the clarity, transparency, and usability of the data.

R4.9: Insufficient discussion of implications for end-user utility

While the absence of Summary of Findings (SoF) tables and EtD frameworks is highlighted, the practical implications for clinicians or decision-makers are not fully elaborated. This connection should be strengthened.

Answer: Thanks for your observation. To address this, we have added a sentence in the Discussion section: “The absence of SoF tables and EtD frameworks in a substantial proportion of CPGs may reduce their practical utility for clinicians and policymakers [20]. SoF tables synthesize the key findings in an accessible format, facilitating transparent and rapid interpretation of the certainty of evidence and magnitude of effects [30]. Likewise, EtD frameworks support decision-makers in adapting recommendations to local contexts by explicitly considering factors such as resource availability, feasibility, equity, and patient preferences [31]. Without these tools, recommendations may lack clarity, contextual relevance, or perceived credibility, potentially limiting their implementation and impact in real-world settings [31].”

20. Guyatt G, Oxman AD, Akl EA, Kunz R, Vist G, Brozek J, et al. GRADE guidelines: 1. Introduction—GRADE evidence profiles and summary of findings tables. J Clin Epidemiol. 2011;64(4):383–94. https://doi.org/10.1016/j.jclinepi.2010.04.026

30. Guyatt GH, Oxman AD, Santesso N, Helfand M, Vist G, Kunz R, et al. GRADE guidelines: 12. Preparing summary of findings tables-binary outcomes. J Clin Epidemiol. 2013;66(2):158-72. https://doi.org/10.1016/j.jclinepi.2012.01.012

31. Alonso-Coello P, Schünemann HJ, Moberg J, Brignardello-Petersen R, Akl EA, Davoli M, et al. GRADE Evidence to Decision (EtD) frame

---

## [Decision Letter · Decision Letter 3]

14 Dec 2025

Methodological characteristics of Peruvian clinical practice guidelines, 2018 - 2023: a scoping review

PONE-D-24-51671R3

Dear Dr. Taype-Rondan,

We’re pleased to inform you that your manuscript has been judged scientifically suitable for publication and will be formally accepted for publication once it meets all outstanding technical requirements.

Kind regards,

Saravana Kumar

Academic Editor

PLOS One

Additional Editor Comments (optional):

Reviewers' comments:

Reviewer's Responses to Questions

**Comments to the Author**

Reviewer #2: All comments have been addressed

2. Is the manuscript technically sound, and do the data support the conclusions?

Reviewer #2: Yes

3. Has the statistical analysis been performed appropriately and rigorously?

Reviewer #2: Yes

4. Have the authors made all data underlying the findings in their manuscript fully available?

Reviewer #2: Yes

5. Is the manuscript presented in an intelligible fashion and written in standard English?

Reviewer #2: Yes

Reviewer #2: This is my third time to review the paper. Overall, I think this paper is interesting and meet the requirement for publication. I still have a little bit humble advice to improve the paper: Consider some data visualization to make the paper more readable and attractive (for example, a heatmap for Table 3, a distribution graph for the level of care or sector of care). Insert Peru-specific data, or explicitly state that such data are unavailable, in Discussion or Limitations part. Just a single word should be enough. The third little comments is to make implications or policy tangible for clinicians. For example, give one brief example where missing EtD hampered implementation. This will make your article more meaningful.

**Do you want your identity to be public for this peer review?** For information about this choice, including consent withdrawal, please see our Privacy Policy

Reviewer #2: No

---

## [Editor Report · Acceptance letter]

PONE-D-24-51671R3

PLOS One

Dear Dr. Taype-Rondan,

I'm pleased to inform you that your manuscript has been deemed suitable for publication in PLOS One. Congratulations! Your manuscript is now being handed over to our production team.

Kind regards,

on behalf of

Professor Saravana Kumar

Academic Editor

PLOS One